Population diversity and relatedness in Sugarbirds (Promeropidae: Promerops spp.)

Haworth Evan S. 1
Cunningham Michael J. michael.cunningham@up.ac.za 1
Calf Tjorve Kathleen M. 2
1 Biochemistry, Genetics & Microbiology, University of Pretoria , Pretoria , South Africa
2 Applied Ecology, Inland Norway University of Applied Sciences , Lillehammer , Norway
Wink Michael
Electronic publication date: 2018 Jun 13
Publication date: 2018
Volume: 6
Electronic Location ID: e5000
Received 2018 Mar 2; Accepted 2018 May 29
Copyright: ©2018 Haworth et al.
Copyright year: 2018
Copyright holder: Haworth et al.
License: This is an open access article distributed under the terms of the Creative Commons Attribution License, which permits unrestricted use, distribution, reproduction and adaptation in any medium and for any purpose provided that it is properly attributed. For attribution, the original author(s), title, publication source (PeerJ) and either DOI or URL of the article must be cited.
License URL: https://creativecommons.org/licenses/by/4.0/

Keywords: Ornithology, Conservation genetics, Sexual selection, Molecular ecology, Phylogeography, Population genetics

Funding: University of Cape Town Funding for this project was received from the Clancy Bequest to the Percy Fitzpatrick Institute for African Ornithology at the University of Cape Town, via an affiliate, Prof. Paulette Bloomer. The funders had no role in study design, data collection and analysis, decision to publish, or preparation of the manuscript.

==============================
Sugarbirds are a family of two socially-monogamous passerine species endemic to southern Africa. Cape and Gurney’s Sugarbird (Promerops cafer and P. gurneyi) differ in abundance, dispersion across their range and in the degree of sexual dimorphism in tail length, factors that affect breeding systems and potentially genetic diversity. According to recent data, P. gurneyi are in decline and revision of the species’ IUCN conservation status to a threatened category may be warranted. It is therefore necessary to understand genetic diversity and risk of inbreeding in this species. We used six polymorphic microsatellite markers and one mitochondrial gene (ND2) to compare genetic diversity in P. cafer from Helderberg Nature Reserve and P. gurneyi from Golden Gate Highlands National Park, sites at the core of each species distribution. We describe novel universal avian primers which amplify the entire ND2 coding sequence across a broad range of bird orders. We observed high mitochondrial and microsatellite diversity in both sugarbird populations, with no detectable inbreeding and large effective population sizes.

Introduction

Sugarbirds (Promeropidae) are a family of two socially-monogamous nectivorous passerine species endemic to southern Africa, with the Cape Sugarbird (Promerops cafer) occurring in the fynbos biome of south-western South Africa and Gurney’s Sugarbird (Promerops gurneyi) occurring in the grasslands of eastern South Africa, Swaziland and Zimbabwe (Fig. 1). Sugarbird occurrence and abundance is closely tied with that of shrubs in the family Proteaceae. The distribution of P. gurneyi is fragmented in comparison to that of P. cafer, owing to the sparse occurrence of the silver sugarbush (Protea roupelliae), a fire sensitive species that is Gurney’s Sugarbird’s preferred source of food, shelter and nesting sites (De Swardt, 1991). By contrast, P. cafer occurs in a region with a much greater diversity, abundance and more even dispersion of Proteaceae shrubs, many of which provide nectar and nest sites for this species. Promerops gurneyi is currently listed as a species of ‘Least Concern’ both globally and within southern Africa (Taylor, Peacock & Wanless, 2015; BirdLife International, 2018). However, recent data presented by Lee, Altwegg & Barnard (2017) suggests that populations of P. gurneyi are in serious decline. The modification of grassland habitat in South Africa, through transformation for agriculture and changes in fire and grazing regimes, has also been implicated in the decline of the Yellow-breasted Pipit (Pietersen et al., 2017), which shares most of its distribution with P. gurneyi.

Figure 1 Sugarbird species distributions and sampling sites.

Distribution of Cape (red) and Gurney’s (green) Sugarbirds. Large circles show sampling sites at Cape Town (lower left) and Golden Gate Highlands National Park (mid right), respectively. The small circle indicates the single sample of P. gurneyi from Sani Pass. The base map of Southern Africa was QGIS while data on Sugarbird occurrence is from the South African Bird Atlas Project 2 (sabap2.adu.org.za) updated in May 2018.

Sugarbirds exhibit fierce territoriality toward other nectivorous birds, including Malachite and Orange-breasted Sunbirds (Nectarinia famosa and Anthobaphes violacea) with which they compete for renewable nectar resources, and toward other Sugarbird individuals with whom they compete for mating opportunities (Daniels, 1987). Taking into account that Proteaceae utilized by P. gurneyi are sparser and less diverse than those utilized by P. cafer (Calf, Downs & Cherry, 2003), and that territory size and quality have effects on breeding success, P. cafer may be substantially less energetically constrained than P. gurneyi in terms of foraging and reproduction (Calf, Downs & Cherry, 2001). Thus, we expect that the abundance and distribution of Proteaceae, such as Protea roupelliae, could have significant effects on seasonal movements, population size and the breeding system of P. gurneyi.

Extremely high rates of extra-pair paternity (EPP) (>70%) have been observed in P. cafer (Henderson, 1999; McFarlane et al., 2009) which is thought to be associated with their extreme sexual dimorphism in tail length, with long-tailed males being more likely to succeed in extra-pair copulations than short-tailed males. Foerster et al. (2003) showed that high rates of EPP promotes increased offspring heterozygosity and fitness in the Blue Tit (Parus caeruleus), which may also be the case for P. cafer, and is consistent with high genetic diversity observed in previous studies (Feldheim, McFarlane & Bowie, 2006). In contrast, the breeding system of P. gurneyi is poorly understood, and rates of EPP have not yet been measured. However, given the low levels of sexual dimorphism in P. gurneyi, we assume that rates of EPP may be lower. Alternatively, male reproductive success in P. gurneyi may be more heavily influenced by other factors, such as resource availability (O’Brien & Dawson, 2011). Given these disparities in occurrence, abundance and breeding system we predict that P. gurneyi populations may show lower genetic diversity than P. cafer, and greater risk of inbreeding within habitat patches.

The ability to analyse genetic markers at a population level has markedly increased our ecological insight and capacity for conservation planning at the intraspecific population level. By contrasting patterns of genetic diversity among populations using rapidly mutating nuclear encoded loci, such as microsatellites, combined with mitochondrial (mtDNA) loci, which are also uniquely sensitive to historical changes in population size due to haploid uniparental inheritance, inferences can be made regarding historical gene-flow (Johnson, Toepfer & Dunn, 2003), historical range expansion (Brito, 2007), and sex-biased dispersal (Melnick & Hoelzer, 1992; Ribeiro et al., 2012). The use of previously identified microsatellites is advantageous as they can be studied using PCR-based fragment scoring methods (Oliveira et al., 2006), which require minimal costs. Additionally, microsatellites are generally highly polymorphic due to mutational instability (Ellegren, 2004), and thus serve as effective population genetic markers for assigning parentage and elucidating population genetic structure.

In this study we aimed to characterize and compare genetic diversity and relatedness in populations of P. gurneyi and P. cafer. We also investigated whether inbreeding poses a risk to the persistence of a population of P. gurneyi in Golden Gate Highlands National Park, within the Maloti-Drakensberg bioregion of southern Africa. We compared this with a population of P. cafer from Helderberg Nature Reserve, near Cape Town, in the south-western Cape. These populations are at the centre of each species distribution and were the sites of previous ecological studies, that provided the samples analysed here. We used six polymorphic microsatellite markers (Feldheim, McFarlane & Bowie, 2006) and designed universal avian primers for the entire mitochondrial NADH dehydrogenase II (ND2) gene to compare levels of heterozygosity, inbreeding, and effective population size between these populations. Our results should serve as a useful basis for future phylogeographic and conservation assessments of P. gurneyi.

Methods and Materials

Sample collection and DNA extraction

All samples were collected during prior ecological studies (Calf, Downs & Cherry, 2001; Calf, Downs & Cherry, 2003). Permits were obtained from the relevant provincial conservation agency (Free State Department of Environmental Affairs and Tourism and CapeNature Permits respectively to Kathleen Calf and Gordon Scholtz, 1998–1999) with mist netting of birds licensed (KMCT) under the South African national bird ringing scheme (http://safring.adu.org.za, permit number HK/P1/02095/001). Blood samples were obtained by brachial vein puncture and capillary capture (approximately 20 µl per bird) from a population of P. gurneyi (n = 48) in Golden Gate Highlands National Park, Free State, in 1998, and from one individual in Sani Pass, KwaZulu-Natal, in 2004. Samples were similarly obtained from a population of P. cafer (n = 63) in Helderberg Nature Reserve, Western Cape, in 1999, where sampling was targeted around nests. Samples were stored in 1mL of a modified PBS solution (phosphate buffered saline blood storage buffer: 3 mM KCl, 8 mM Na2HPO4, 2 mM KH2PO4, 0.14 M NaCl, 6 mM EDTA, 0.2% NaN3 (w/v)) which was kept on ice after sampling and stored long-term at −20 °C. This buffer includes EDTA to prevent enzymatic digestion of nucleic acids, while the Sodium Azide, NaN3, is an antimicrobial.

Samples from other species, used to determine the taxonomic range of ND2 primer amplification, were available from previous studies in our laboratory. These included mountain pipit, Anthus hoeschi, blue crane, Anthropoides paradiseus, Angolan cave chat, Cossypha ansorgei, Cape parrot, Poicephalus robustus, Namaqua sandgrouse, Pterocles namaqua, and eastern clapper lark, Mirafra fasciolata. Tissue samples of chicken, Gallus gallus domesticus and Ostrich, Struthio camelus, were obtained from supermarket meat, and marsh owl, Asio capensis, from a roadkilled individual sampled from a carcass found beside the R50 road at Delmas (Long, Lat: 028.681, −26.141).

We extracted genomic DNA by salting out. Digests were prepared by adding 1 mg blood to 500 µL DNA lysis buffer (100 mM NaCl, 50 mM Tris.HCl, 100 mM EDTA, 1% SDS w/v) supplemented with 0.2 mg proteinase-K, before incubation at 57 °C overnight. Following digestion, a volume of 20 µL RNase A (25 mg/mL) was added, before incubation at 37 °C for 60 min. Proteins and other cellular contents were precipitated by addition of 180 µL 5 M NaCl (final concentration 1.3 M) followed by agitation and centrifugation. DNA was then extracted by combining the supernatant with ice-cold isopropyl alcohol (1:1), washed with 70% ethanol, and resuspended in 12 × TE buffer (5 mM tris, 0.5 mM EDTA). We confirmed successful extraction using agarose gel electrophoresis (1% agarose, 1 × TAE buffer, 100 V, 15 min), assessed DNA quantity and quality using a NanoDrop™ spectrophotometer, and prepared working stocks of 50 ng/µL.

Microsatellite genotyping

We used multiplex PCR to amplify the six microsatellite loci developed by Feldheim, McFarlane & Bowie (2006). Reactions contained 5 µL Platinum® PCR Multiplex Mix (Applied Biosystems, Foster City, CA, USA), 25 ng template DNA, 0.015 M of each primer (Table 1), made up to a final volume of 10 µL using ultra-pure H2O. Reactions were amplified in an Applied Biosystems™ 2720 thermocycler under the following conditions: long denaturation at 95 °C for 10 min followed by 33 cycles of denaturation at 95 °C for 30 s, annealing at 56 °C for 1 min 30 s, and extension at 72 °C for 30 s, with a final extension step at 60 °C for 30 min.

Table 1 Microsatellite and mitochondrial primer sequences.

Characterization of sugarbird microsatellite and mitochondrial markers. The motif repeat type, primer sequence and fluorescent label are shown.

Locus	Repeat type	Primer sequence 5′ to 3′	Reference	
Pro24	Hexa	F: TCGTCATCTTGCAACCAAAA (FAM)	Feldheim, McFarlane & Bowie (2006)	
R: TCAGCAGCAAACATGAAACC		
Pro25	Tetra	F: CGAGAGCCAGGATTCATTTTCC (VIC)	Feldheim, McFarlane & Bowie (2006)	
R: AGCCAGAATTTGTCCTGTCTG		
Pro66	Tetra	F: GCTTGATTAAGGTGCCGAAA (NED)	Feldheim, McFarlane & Bowie (2006)	
R: GCAGGACACAGAGCACTCAA		
Pro86	Penta	F: CAGACCTTGGAACAGGCTTC (VIC)	Feldheim, McFarlane & Bowie (2006)	
R: GGCTCCCTCAATTCCTTCTC		
Pro19	Tetra	F: TGGAACAGTCCACTTCATGC (NED)	Feldheim, McFarlane & Bowie (2006)	
R: CAACTTTCCTAGCAAAAGGCAC		
Pro90	Tetra	F: TTGGAGGGAAGAAGATCTGGG (PET)	Feldheim, McFarlane & Bowie (2006)	
R: CATTCCTTGCCCATTCTGCTG		
ND2		F(L3977): GGCCCATACCCCGAAAATGA	(This study)	
	R(H5191): GGATCGAAGCCCATCTGCCTA		

We used agarose gel electrophoresis to confirm successful PCR amplification (2% agarose, 1 × TAE buffer, 100 V, 15 min), and conducted fragment analysis on an ABI3500xl Genetic Analyzer with a Liz-500 size standard (Applied Biosystems) at the DNA Sanger Sequencing Facility, University of Pretoria. We analysed electropherogram results using GeneMarker® v1.95 (SoftGenetics, State College, PA, USA) and designed separate genotyping panels for each Sugarbird species. We checked the consistency of fragment assignment by replicate amplification and scoring of 20% of samples.

ND2 primer design

Sorenson et al. (1999) developed a range of universal avian mitochondrial primers, however, most of these make use of degenerate sites which reduce their utility for sequencing. We used the Primer3 v.2.3.7 (Untergasser et al., 2012) plugin in Geneious® vR10.2.2 to design universal avian ND2 primers against Taeniopygia guttata (NC_007897.1) with comparative alignment across 16 other bird species (Table 2). Primer L3977 is within the Methionine tRNA gene (positions 3958–3977 in T. guttata). Primer H5191 is within the Asparagine tRNA gene (positions complementing 5211–5191 in T. guttata), resulting in an expected product length of 1254 bp (1213 bp target sequence, including 23 bp of the Methionine tRNA, the complete 1,041 bp of ND2, 70 bp Tryptophan tRNA gene, 69 bp complementary to the Alanine tRNA and 10 bp non-genic nucleotides). We authenticated our primers by PCR amplification across 11 species, spanning the avian tree of life (Barker et al., 2004; Jetz et al., 2012; Prum et al., 2017) and used NCBI BLAST (Altchul et al., 1990) to confirm successful gene-targeting for a subset of individuals following Sanger sequencing. By designing primers in the conserved tRNAs flanking ND2, we were able to amplify the entire ND2 coding sequence across a broad range of species using a single protocol.

Table 2 ND2 primer site alignment across Avian orders.

Alignment of ND2 primer sequences across 17 bird species spanning several orders. Genbank accession numbers for sequences used in primer design are shown at left. L3977 in tRNA-Met matches perfectly to all species excepting a 1bp mid-primer A-G transition in Aythya americana. H5191 matches perfectly to Taeniopygia guttata (the design reference) with 1–3 mid-primer mismatches in all other species.

NCBI accession number	Species	L3977	H5191	
		G	G	C	C	C	A	T	A	C	C	C	C	G	A	A	A	A	T	G	A	G	G	A	T	C	G	A	A	G	C	C	C	A	T	C	T	G	C	C	T	A	
NC_007897.1	Zebra finch (Taeniopygia guttata)	•	•	•	•	•	•	•	•	•	•	•	•	•	•	•	•	•	•	•	•	•	•	•	•	•	•	•	•	•	•	•	•	•	•	•	•	•	•	•	•	•	
NC_000880.1	Village indigobird (Vidua chalybeata)	•	•	•	•	•	•	•	•	•	•	•	•	•	•	•	•	•	•	•	•	•	•	•	•	•	•	•	•	•	•	•	•	•	•	•	•	T	•	•	•	•	
NC_010774.1	Mrs Hume’s pheasant (Syrmaticus humiae)	•	•	•	•	•	•	•	•	•	•	•	•	•	•	•	•	•	•	•	•	•	•	•	•	•	•	•	C	•	•	•	•	•	•	•	•	•	•	•	•	•	
NC_010771.1	Elliot’s pheasant (Syrmaticus ellioti	•	•	•	•	•	•	•	•	•	•	•	•	•	•	•	•	•	•	•	•	•	•	•	•	•	•	•	C	•	•	•	•	•	•	•	•	•	•	•	•	•	
NC_010770.1	Reeves’s pheasant (Syrmaticus reevesii)	•	•	•	•	•	•	•	•	•	•	•	•	•	•	•	•	•	•	•	•	•	•	•	•	•	•	•	C	•	•	•	•	•	•	•	•	•	•	•	•	•	
NC_010767.1	Copper pheasant (Syrmaticus soemmerringi)	•	•	•	•	•	•	•	•	•	•	•	•	•	•	•	•	•	•	•	•	•	•	•	•	•	•	•	C	•	•	•	•	•	•	•	•	•	A	•	•	•	
NC_010781.1	Crested fireback (Lophura ignita)	•	•	•	•	•	•	•	•	•	•	•	•	•	•	•	•	•	•	•	•	•	•	•	•	•	•	•	C	•	•	•	•	•	•	•	•	•	•	•	•	•	
NC_010778.1	Green pheasant (Phasianus versicolor)	•	•	•	•	•	•	•	•	•	•	•	•	•	•	•	•	•	•	•	•	•	•	•	•	•	•	•	C	•	•	•	•	•	•	•	•	•	•	•	•	•	
NC_007238.1	Green junglefowl (Gallus varius)	•	•	•	•	•	•	•	•	•	•	•	•	•	•	•	•	•	•	•	•	•	•	•	•	•	T	•	•	•	•	•	•	•	•	•	•	•	•	•	•	•	
NC_007240.1	Grey junglefowl (Gallus sonneratii)	•	•	•	•	•	•	•	•	•	•	•	•	•	•	•	•	•	•	•	•	•	•	•	•	•	•	•	•	•	•	•	•	•	•	•	•	•	A	•	•	•	
NC_007329.1	Sri Lankan junglefowl (Gallus lafayetii)	•	•	•	•	•	•	•	•	•	•	•	•	•	•	•	•	•	•	•	•	•	•	•	•	•	•	•	•	•	•	•	•	•	•	•	•	•	A	•	•	•	
NC_003408.1	Japanese quail (Coturnix japonica)	•	•	•	•	•	•	•	•	•	•	•	•	•	•	•	•	•	•	•	•	•	•	•	•	•	•	•	•	•	•	•	•	•	•	•	•	•	A	•	•	•	
NC_010195.1	Domesticated turkey (Meleagris gallopavo)	•	•	•	•	•	•	•	•	•	•	•	•	•	•	•	•	•	•	•	•	•	•	•	•	•	T	•	C	•	•	•	•	•	•	•	•	•	•	•	•	•	
NC_000877.1	Redhead (Aythya americana)	•	•	•	•	•	•	•	•	•	•	•	•	•	G	•	•	•	•	•	•	•	•	•	•	•	•	•	•	•	•	•	•	C	•	•	•	•	A	•	•	•	
NC_029846.1	Lesser kestrel (Falco naumanni)	•	•	•	•	•	•	•	•	•	•	•	•	•	•	•	•	•	•	•	•	•	•	•	•	•	•	•	C	•	•	•	•	•	•	A	•	•	A	•	•	•	
NC_000878.1	Peregrine falcon (Falco peregrinus)	•	•	•	•	•	•	•	•	•	•	•	•	•	•	•	•	•	•	•	•	•	•	•	•	•	•	•	C	•	•	•	•	•	•	•	•	•	A	•	•	•	
NC_000879.1	Grey-headed broadbill (Smithornis sharpei)	•	•	•	•	•	•	•	•	•	•	•	•	•	•	•	•	•	•	•	•	•	•	•	•	•	•	•	•	•	•	•	•	•	•	•	•	•	A	•	•	•	

Mitochondrial gene sequencing

Mitochondrial genes were amplified in 10 µL PCR reactions comprised of 1X PCR buffer, 1.5 mM MgCl2, 0.2 mM dNTPs, 0.4 µM forward and reverse primer (Table 1), 0.5 U/µL Supertherm® Taq polymerase, and 2 µL of template DNA, made up to final volume with ultra-pure H2O. Reactions were placed into an Applied Biosystems™ 2720 thermocycler for amplification under the following conditions: long denaturation at 94 °C for 5 min, followed by 33 cycles of denaturation at 94 °C for 30 s, annealing at 56 °C for 30 s, and extension at 72 °C for 45 s, followed by 2 cycles of denaturation at 94 °C for 30 s and extension at 72 °C for 1 min 30 s. Successful amplification was confirmed using agarose gel electrophoresis (2% agarose, 100 V, 15 min), and DNA precipitation was performed to remove unincorporated dNTPs and primers in preparation for sequencing.

Cycle-sequencing reactions each comprised of 1 µL BigDye® Terminator v3.1 Ready Reaction Mix, 1 µL BigDye® Sequencing Buffer, 3.2 µM forward primer (Table 1), and 2 µL template DNA, made up to a final volume of 10 µL with ultra-pure H2O. Reactions were placed into an Applied Biosystems™ 2720 thermocycler for amplification under the following conditions: long denaturation at 96 °C for 3 min, followed by 30 cycles of denaturation at 96 °C, and annealing and extension at 60 °C for 4 min. Following DNA precipitation, cycle-sequencing products were sequenced on either ABI3730xl or ABI3500xl DNA Analysers at the Sanger Sequencing Facility, University of Pretoria. The process was then repeated using the reverse primer to confirm sequences and increase coverage.

Data analysis

We used Cervus 3.0.7 (Kalinowski, Taper & Marshall, 2007) to measure allele frequencies at the six microsatellite loci, Genepop v4.6 (Rousset, 2008) to test for deviations from Hardy-Weinberg and linkage disequilibrium among loci, and Micro-checker v2.2.3 (Van Oosterhout et al., 2004) to identify and correct for null-alleles. We conducted parentage analysis using Cervus 3.0.7 (Kalinowski, Taper & Marshall, 2007) in order to identify parent–offspring trios. We conducted parent-pair simulations for 10,000 offspring under the conservative assumption that 25% of potential fathers had been sampled after trial runs with different parameters, allowing two mismatches to account for the Z-linked locus (Pro86) as well as any null-alleles or misscoring.

We used Coancestry V1.0.1.7 (Wang, 2011) to calculate pairwise relatedness estimates (Queller & Goodnight, 1989) and inbreeding coefficients (Lynch & Ritland, 1999) for both sugarbird populations, and used R-Studio V1.1.383 (Ross et al., 1996) to graph the frequency distribution of pairwise relatedness estimates. We then estimated the effective size of both populations using the linkage-disequilibrium method (Waples & Do, 2010) implemented in NEEstimator V2.01 (Do et al., 2014) with jack-knifing across loci. Given that P. cafer exhibits high rates of EPP and given that rates of EPP have not yet been measured in P. gurneyi, we estimated effective population sizes assuming both random and monogamous mating.

Sampling of P. cafer was targeted around nests, and thus our P. cafer dataset included a higher proportion of close relatives (parent–offspring, siblings, and half-sibling). In order to investigate whether sampling methods may affect our estimates of effective population size, we used Friends and Family (De Jager et al., 2017) to produce a reduced P. cafer dataset with a lower overall mean relatedness. This analysis identified groups of potentially related individuals (relatedness > 0.25), and randomly removed individuals from each group until sample size was equal for both datasets.

We used MEGA 7.0.26 (Kumar, Stecher & Tamura, 2016) to align and trim our mitochondrial sequence data and to calculate nucleotide diversity. We then converted our sequence data to NEXUS format using PGDSpider v2.1.1.2 (Lischer & Excoffier, 2012), and used Popart v1.7 (Bandelt, Forster & Röhl, 1999) to construct a median-joining ND2 haplotype network. In addition to the sequence data that we produced, complete ND2 coding sequences of both P. cafer (accession number DQ125990) and P. gurneyi (accession number GU16832.1) were downloaded from GenBank and included in our analyses.

Results

Microsatellite diversity

All six microsatellite loci were highly polymorphic in both P. cafer and P. gurneyi (Table 3). There were no significant deviations from Hardy-Weinberg equilibrium within loci but there was a general tendency towards a slight deficiency of heterozygotes across loci (10 of 12 comparisons). Numbers of alleles and heterozygosity were slightly but not significantly higher in P. cafer than P. gurneyi (paired difference t-test t = 1.2, df = 5, N.S.). We suspected null-alleles at the marker Pro66 due to excess homozygosity and failed amplification of this locus in one sample, and so we used Micro-checker to obtain corrected genotypes at this locus.

Table 3 Genetic diversity in Cape and Gurney’s Sugarbirds. Diversity statistics in P. gurneyi and P. cafer.

Statistics for microsatellite loci are Number of alleles (NA), Sample size (n), Observed heterozygosity (Ho), Expected heterozygosity (He), Polymorphic Information Content (PIC), Locus inbreeding coefficient—potentially representing observed homozygote excess due to null alleles (F(null)) and Allele size ranges. Mitochondrial statistics also include nucleotide diversity (pi) and gene-diversity (H, equivalent to He).

Locus	NA	n	Ho	He	PIC	F(null)	Size range (bp)	
P. gurneyi	
Pro24	14	49	0.82	0.84	0.81	0.010	181–262	
Pro25	9	49	0.80	0.81	0.78	0.003	201–235	
Pro86	9	24	0.83	0.84	0.80	−0.006	302–352	
Pro19	10	49	0.78	0.85	0.82	0.042	166–203	
Pro66	11	31	0.71	0.88	0.86	0.096	319–355	
Pro90	12	49	0.90	0.87	0.85	−0.019	204–241	
Mean	10.5	–	0.81	0.85	0.82			
Locus	NA	n	π	H	Size (bp)			
ND2	6	15	0.0012	0.76	1,041			
P. cafer	
Pro24	19	63	0.86	0.92	0.90	0.029	187–324	
Pro25	16	63	0.84	0.83	0.81	0.012	185–278	
Pro86	16	37	0.81	0.89	0.87	0.040	273–352	
Pro19	8	63	0.76	0.79	0.75	0.006	154–186	
Pro66	20	61	0.84	0.92	0.91	0.046	275–340	
Pro90	15	63	0.87	0.88	0.87	0.003	180–245	
Mean	15.2	–	0.83	0.87	0.85			
Locus	NA	n	π	H	Size (bp)			
ND2	8	11	0.0024	0.84	1,041			

Additionally, 1–2 base pair shifts were observed at several Pro66 alleles (4 bp repeat motif), these scored consistently across replicate PCRs and parent–offspring comparisons and were considered as separate alleles in analyses. We also observed a slight overlap in allele size ranges between markers Pro25 and Pro86 in the P. cafer panel, both of which are labelled with the same fluorophore. Fortunately, most individuals with alleles in the overlapping region were identified as females, which are hemizygous for the Z-linked Pro86, and so manual scoring was possible.

ND2 primer authentication

As shown in Table 2, the region to which the forward primer L3977 binds is highly conserved across all 17-species included in our alignment, with only one variable site. The region to which H5191 binds is less conserved, with six variable sites across 17 species, however, all 11 of the bird species included produced clean bands following PCR amplification with the avian ND2 primers L3977 and H5191. Of the 5-additional species selected for Sanger sequencing (A. hoeschi, A. paradiseus, C. ansorgei, S. camelus, and P. robustus), each sample yielded over 1100bp of high quality sequence data (GenBank accession numbers; MG972851, MG972852, MG972853, MG972854, and MG972855).

Mitochondrial diversity

We obtained 1,150 bp, including the complete ND2 coding sequence, from 10 P. cafer (GenBank accession numbers; MG972856 –MG972865) and 15 P. gurneyi (GenBank accession numbers; MG972866 –MG972879). These were combined for phylogenetic analysis with the single existing ND2 sequences of each species from GenBank, each of which was a singleton haplotype. MtDNA diversity matches the pattern from microsatellites, with higher nucleotide and haplotype diversity observed in P. cafer (π = 0.0024; H = 0.84) compared to P. gurneyi (π = 0.0012; H = 0.76), despite a smaller sample of the former. We observed eight haplotypes in P. cafer and 6 haplotypes in P. gurneyi (Fig. 2).

Figure 2 ND2 haplotype networks for Cape and Gurney’s Sugarbirds.

Mean uncorrected sequence diversity is 0.24% in P. cafer (maximum 0.6%) and 0.12% in P. gurneyi (maximum 0.3%). Net divergence between species is 2.70% (28 nucleotide differences), ranging from 2.70 to 3.07% (28–31 differences).

Inbreeding and relatedness

Mean relatedness and inbreeding coefficients were low for both species. Bell-shaped curves were obtained for both relatedness frequency distributions (Fig. 3), with mean relatedness being slightly lower in P. gurneyi (mean  = − 0.022; variance = 0.040) than in P. cafer (mean  = − 0.016; variance = 0.030) with wide variance. Mean inbreeding coefficients were slightly higher for P. gurneyi (mean F = 0.028; variance = 0.026) than for P. cafer (mean F = 0.024; variance = 0.013). We also used Cervus (Kalinowski, Taper & Marshall, 2007) to confirm four parent–offspring pairs in our P. gurneyi dataset and six parent–offspring pairs in our P. cafer dataset, although we suspect that the proportion of second order relatives is significantly higher in our P. cafer dataset due to sampling of some half-sibling fledgelings.

Figure 3 Distribution of relatedness estimates in Cape and Gurney’s Sugarbirds.

Histograms of Queller & Goodnight’s (1989) relatedness statistic among (A) 49 Gurney’s sugarbirds from Golden Gate Highlands National Park and (B) 63 Cape Sugarbirds from Helderberg Nature Reserve. Columns indicate the rescaled density of pairwise relatedness values in each population, red lines are smoothed curves matching the shape of each distribution. This measure varies from −1 to 1, with 0 representing Hardy-Weinberg expected similarity of multilocus genotypes, negative values indicating genotypes that are less similar than expected from random resampling the data, positive values show genotypes that are more similar than expected under random resampling.

Effective population size

The effective population size of P. cafer in Helderberg, Western Cape, was estimated to be 99 individuals (95% CI [66–182]) assuming random mating and 198 individuals (95% CI [132–356]) assuming monogamous mating. The effective population size of P. gurneyi in Golden Gate Highlands National Park, Free State, was estimated to be 133 individuals (95% CI [55–983]) assuming random mating and 223 individuals (95% CI [111–1,627]) assuming monogamous mating. The reduced P. cafer dataset, filtered for relatives, yielded somewhat higher estimates of effective population size, of 157 individuals (95% CI [87–582]) assuming random mating, and 316 individuals (95% CI [175–1,144]) assuming monogamous mating.

Discussion

Both sugarbird species exhibit high levels of microsatellite and mitochondrial diversity, with relatively large local effective population sizes and no detectable inbreeding. This is consistent with levels of diversity previously observed in P. cafer (Feldheim, McFarlane & Bowie, 2006) and other African nectivorous birds, such as orange-breasted sunbird (Chan, Van Vuuren & Cherry, 2011), Ruwenzori double-collared sunbird (Cinnyris stuhlmanni) (Bowie, Sellas & Feldheim, 2010) and white-eyes (Zosterops sp.) (Oatley et al., 2017).

High diversity in P. gurneyi was unexpected, given their fragmented occurrence and the sparse distribution of suitable habitat. Seasonal migration of P. gurneyi has been inferred by bird monitoring projects, such as the South African Bird Atlas Project 2—(SABAP2) citizen science project. These movements include altitudinal movements within regions, and movement from the inland escarpment to the south-east coast (De Swardt, 1991; Hockey, Dean & Ryan, 2005; SABAP2, 2017). One possibility is that diversity in P. gurneyi is maintained through unrecognised gene-flow among regional populations, which are separated by large areas of unsuitable habitat. Alternatively, high levels of diversity may reflect a historically large metapopulation of P. gurneyi within this particular region. Our study site, Golden Gate Highlands National Park is one of several large protected areas within the Maloti-Drakensberg Bioregion, the largest untransformed area within the distribution of P. gurneyi. Stands of Protea roupelliae, a key feature of this species habitat, are dispersed across this bioregion, with intervening Protea caffra shrubs and other nectar providing plants providing connectivity among patches. SABAP2 results suggest that the Maloti-Drakensberg area is the stronghold of this species, with the possibility of more extensive connections to the south-eastern coastal area. The single sample from Sani Pass, 130 km SE of Golden Gate, in the Ukhahlamba-Drakensberg World Heritage area, shared a mitochondrial haplotype with the Golden Gate population but also carried several unique microsatellite alleles suggesting the possibility of population divergence across this region but not long-term isolation.

Thus our samples from the north-eastern periphery of a large and well connected regional population may not reflect diversity in other isolated regions further North in Mpumalanga and Limpopo provinces of South Africa, and the eastern highlands of Manicaland province, Zimbabwe. We expect that the inclusion of samples from these regions would yield additional mitochondrial haplotypes and private microsatellite alleles. Irrespective of the contribution of phylogeographic structure to this diversity, it is unlikely that inbreeding poses an immediate risk to the persistence of P. gurneyi in Golden Gate Highlands National Park.

Differences in genetic diversity

All measures of diversity were higher in P. cafer than in P. gurneyi. Several factors could contribute to this disparity. Firstly, our sample of P. cafer was somewhat larger than that of P. gurneyi. This is unlikely to be a substantial contributor to interspecific differences in diversity as substantial samples (>48 individuals) were analysed from each species, each from a single population at the centre of the species range. Some of our measures, such as heterozygosity and nucleotide diversity, are robust to variation in sample size (Nei, 1987; Pruett & Winker, 2008; Hale, Burg & Steeves, 2012). Although the proportion of close relatives was low in both datasets, this was higher in P. cafer, which would tend to reduce observed diversity in that species, contrary to the observed differences. Additionally, these differences in diversity remained after randomly subsampling P. cafer to compare equal sized samples.

Secondly, there may potentially be some degree of ascertainment bias—we used microsatellite markers which were developed in P. cafer and selected for high allelic diversity (Feldheim, McFarlane & Bowie, 2006); therefore, there may be lower allelic diversity at these markers in P. gurneyi due to increased mutational stability (Huang et al., 2016). All microsatellite markers showed high diversity in both species, with extensive overlap in allele size ranges and a low and similar frequency of null alleles. Ascertainment bias would not affect differences in mtDNA diversity.

Diversity may be influenced by the promiscuous breeding system in P. cafer, which shows a high level of extra-pair paternity despite long-term social monogamy (Henderson, 1998; McFarlane et al., 2009. Promerops gurneyi also shows long-term social monogamy, however the frequency of extra-pair paternity may be lower in this species, where territories are more dispersed and reduced sexual dimorphism implies lower levels of sexual selection than in the extremely dimorphic P. cafer. We believe that the best explanation for higher diversity in P. cafer may be higher abundance and higher connectivity across the species’ distribution, in the Proteaceae rich fynbos biome.

Inbreeding and relatedness

Our panel only included six microsatellite markers, but all were highly polymorphic and effective in assigning parentage. The frequency distribution of relatedness estimates yielded bell-shaped curves for both populations (Fig. 3), suggesting that our sampling was not biased by related individuals. Mean relatedness was low for both populations, and slightly higher for P. cafer (−0.016) compared to P. gurneyi (−0.022). Mean inbreeding coefficients were low in both populations, with little variation in the degree of inbreeding among individuals. We detected slightly higher rates of inbreeding in P. gurneyi (mean F = 0.028) compared to P. cafer (mean F = 0.024), which may be attributed to higher overall diversity in P. cafer. In sum, we conclude that it is highly unlikely that inbreeding poses a threat to the persistence of either P. gurneyi in the Maloti-Drakensberg or P. cafer in the south-western Cape. However, it remains to be seen whether the same can be said of the several smaller disjunct regional populations of P. gurneyi.

Effective population size

Sugarbirds form socially-monogamous breeding pairs which often re-unite in subsequent breeding seasons, however, while P. cafer exhibits high rates of EPP (>70%), the breeding system of P. gurneyi remains poorly understood (Henderson, 1999; Calf, Downs & Cherry, 2001; Hockey, Dean & Ryan, 2005). We predict that due to competition for territories and sexual selection the mating system of Sugarbirds is not random, nor is it purely monogamous. The linkage disequilibrium method implemented in NEEstimator V2.01 requires specification of either a random or monogamous mating system, we estimated the effective size of both populations assuming either random or monogamous mating.

The effective size estimated for each population was considerably larger than our sample size, suggesting that both populations were derived from relatively large and stable historical populations. The effective population size of P. cafer was estimated as 99 individuals (95% CI [66–182]) under the assumption of random mating, and 198 individuals (95% CI [132–356]) under the assumption of monogamous mating. The effective population size of P. gurneyi was estimated as 133 individuals (95% CI [55–983]) under the assumption of random mating, and 223 individuals (95% CI [111–1,627]) under the assumption of monogamous mating. Estimates of effective size based on linkage are upwardly biased by sample size, with confidence intervals strongly influenced by the number of loci analysed. Estimates of effective population size were larger for P. gurneyi than for P. cafer, perhaps due to the slightly higher proportion of relatives in the P. cafer dataset. There are wide and broadly overlapping confidence intervals around all these estimates. These figures may be understood as indices of local abundance and relatedness, with values larger than sample size suggesting that additional sampling is required to attain stable estimates of long-term effective population size. In contrast to sampling of P. gurneyi, sampling of P. cafer was targeted around nests, and therefore contained a slightly higher proportion of relatives. Filtering the P. cafer dataset for relatives yielded a much higher estimate of effective population size, with 95% upper confidence intervals comparable to those estimated for P. gurneyi. This demonstrates that slight differences in sampling strategies can affect interspecific comparisons of effective population size, specifically when using methods based on linkage-disequilibrium. In the future it may be informative to use coalescent-based Bayesian approaches, which are independent of sample size but generally give wide confidence intervals for microsatellite data and may demand a better understanding of population structure and gene-flow within species.

Conclusions

We observed unexpectedly high microsatellite and mitochondrial diversity in P. gurneyi with no detectable inbreeding. It remains unclear whether differences in genetic diversity between P. gurneyi and P. cafer reflect differences in breeding systems or connectivity among regional populations. In future studies we hope to determine whether phylogeographic structure exists between the disjunct regional populations of P. gurneyi in southern Africa; however, sample collection is made difficult by the sparse distribution of P. roupelliae and the unpredictable seasonal movements of this species in relation to weather and flowering feed trees. We also intend to increase our panel of nuclear markers by including universal avian microsatellites (Dawson et al., 2013) and to extend our comparative assessment to similarly fragmented declining grassland species such as the Drakensberg Rockjumper, the Sentinel Rock Thrush, and the Ground Woodpecker.

Supplemental Information

Supplemental Information 1 Fasta alignment of sequences validating universal Avian ND2 primers L3977–H5191

Click here for additional data file.

Supplemental Information 2 Fasta alignment of ND2 sequences from Cape and Gurney’s Sugarbirds

Click here for additional data file.

Supplemental Information 3 Microsatellite genotype data for Gurney’s Sugarbird

Click here for additional data file.

Supplemental Information 4 Microsatellite genotype data for Cape Sugarbird

Click here for additional data file.

Supplemental Information 5 R code for relatedness histograms

Click here for additional data file.

We would like to acknowledge Craig Symes and Bradley Gibbons for their assistance with sample collection, and Paulette Bloomer, Arrie Klopper, Claire Lenahan, Kate Henderson and Dawie de Swardt for fruitful discussions that have informed this article. We dedicate this article to the memory of Matthew Haworth.

Additional Information and Declarations

Competing Interests

Author Contributions

Animal Ethics

DNA Deposition

Data Availability

The authors declare there are no competing interests.

Evan S. Haworth conceived and designed the experiments, performed the experiments, analyzed the data, prepared figures and/or tables, authored or reviewed drafts of the paper, approved the final draft.

Michael J. Cunningham conceived and designed the experiments, analyzed the data, contributed reagents/materials/analysis tools, prepared figures and/or tables, authored or reviewed drafts of the paper, approved the final draft.

Kathleen M. Calf Tjorve conceived and designed the experiments, contributed reagents/materials/analysis tools, sample collection, Ecological Data.

The following information was supplied relating to ethical approvals (i.e., approving body and any reference numbers):

All blood samples were for this study were collected for prior ecological studies for an MSc study by Kath Calf (Tjorve) at Stellenbosch University (1998–1999) under the supervision of Prof. Michael Cherry (Stellenbosch University) and Prof. Colleen Downs (University of KZN)(detailed in publications Calf, Downs & Cherry 2001, 2003). Birds were captured in mist-nets under licence from SAFRING South Africa (Kathleen Calf), and permits from Western Cape Province and Free State Province (permit number HK/P1/02095/001). All sampled birds were released without injury and included in subsequent observational behavioural studies.

The following information was supplied regarding the deposition of DNA sequences:

GenBank accession numbers: MG972851 to MG972879

The following information was supplied regarding data availability:

The raw data and code are provided in the Supplemental Files.

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
