# Peer review of "Population diversity and relatedness in Sugarbirds (Promeropidae: Promerops spp.)"

_PeerJ, doi:10.7717/peerj.5000_

## Round 0.1 · original submission · Minor Revisions

Dear authors

Our reviewers found a few items which need revision. We look forward to receive your revision in time.

Kind regards

Michael Wink
Academic editor

·

Basic reporting

no comment

Experimental design

no comment

Validity of the findings

These seem appropriate, although I would like to see some statistical comparisons between the diversity metrics of the two species as I expect they are not significantly different.

Additional comments

This is a straightforward paper that compares diversity measures from one population of each sugarbird species. A secondary contribution, are the publication of two new mtDNA primers that appear to be of broad utility to PCR-amplify the NADH2 locus across birds. I have only relatively minor comments that I hope the authors will find useful in revising their paper.

Introduction
1) I would move the first paragraph to the end of the intro. First lead with the information pertaining to sugarbirds, then on to the markers to be used and approaches to be adopted.
2) Line 5 – make mention that not only does mtDNA have a fast mutation rate, but it has a faster coalescence time relative to nuclear DNA due to its smaller effective population size.
3) Line 7 – a more bird specific ref may be useful, for a southern African example see Ribeiro et al. 2012 Mol Ecol 21: 662-672.
4) Line 11 – hanging sentence “… and thus serve as effective population genetic markers…FOR WHAT…”
5) Lines 28 & 29 – provide latin names for the two sunbird species mentioned.
6) Line 30 – the use of the wording “mating opportunities” in the sentence structure is confusing as it implies that sugarbirds and sunbirds are competing for mating opportunities – i.e. with each other, which is certainly not the case.
7) Line 35 – Proteaceae should not be in italics
8) Line 38 – language “…, and IT is thought…”

Methods
1) Details of the Animal Use Committee (IACUC) permissions with reference to a protocol number should be provided.
2) Lines 64, 66 and various other places in the manuscript – I do not think Free State or Western Cape should be hyphenated.
3) Line 69 – Cape Nature Conservation is now called CapeNature
4) Line 120 – what is “Noah’s Ark”? This seems unnecessary. Why not just say xx species, spanning a large extent of avian diversity and then cite a general bird phylogenetics paper such as Hackett et al. 2008 Science, Prum et al. 2015 Nature.
5) Line 159 – I do not think Jack-knifing is should be in caps.

Results
1) Line 95 – drop the “Noah’s ark”
2) Drop Figure 2, it does not add to the paper.
3) Line 218 – full stop is missing.

Discussion
1) line 233 – add latin name for sunbird
2) lines 230-233 – you could extend this section by adding data on Zosterops species as another nectivorous bird – see papers by Graeme Oatley for southern African species.
3) line 235 – “…inferred BY bird…
4) Line 230 – make clear that you are talking about genetic diversity
5) Line 261 – measures of diversity – can these be compared statistically. Many of the values are very similar and I expect that they do not differ significantly, which would suggest that in the core of their respective ranges the two sugarbird species retain similar levels of genetic diversity.
6) Line 266 – provide a ref in support of the statement that these measures are “robust to variation in sample size”.
7) Lines 302-304 – provide refs to support these statements
8) Lines 312-315 – remove, this is repetition of the results
9) Line 329 – extent the sentence to say why Bayesian approaches were not adopted – nature of the data?

Figures and tables

Figure 1 – be more specific than Cape Town. Provide co-ordinates in the figure legend.

Figure 2 – can be removed from the paper.

Figure 3 – Specify what kind of network this is in the legend. In addition add the percentage uncorrected sequence divergence to the figure above the cross-ticks separating the two species.

Table 3 – Add species common names to the figure.

Rauri C. K. Bowie
March 2018
[email protected]

Reviewer 2 ·

Basic reporting

The article is clear, unambiguous, and the text is technically correct. The article does include sufficient introduction and background of how the work fits into the broader field of knowledge. The work is also appropriately referenced and figures and tables are easy to follow.
In summary, two species of sugarbird are found in different geographical regions, where one Promerops cafer is in a region with much greater diversity than the other P. gurneyi. The authors therefore propose that the latter may have lower genetic diversity and greater inbreeding so the objective is to provide phylogeographic and conservation assessments for P. gurneyi because it may be on the decline. To do this the authors make use of two genetic markers.

Experimental design

The submission describes original primary research which builds on previous ecological studies of the very same samples. The knowledge gap being investigated is identified, as the authors cover the differences in genetic diversity, inbreeding and relatedness, and effective population size in good detail and the study contributes to filling that gap.
The investigation has been conducted rigorously: For the microsatellite genotyping the authors checked consistency of fragment assignment by replicate amplification. Although only six microsatellite loci were used, they were all highly polymorphic and effective in assigning parentage. The ND2 primer design and authentication across a “Noah’s Ark” of birds is a neat addition to the paper. The mitochondrial gene sequencing was performed using both forward and reverse primers to confirm sequences and increase coverage.
All the methods are described with sufficient information to be reproducible by another investigator.
The research was conducted in conformity with prevailing ethical standards in the field and permits were obtained from the relevant provincial conservation agencies.

I could not find the GenBank accession numbers: MG972851 to MG972879, however, they are probably embargoed until publication.

Validity of the findings

The data on which the conclusions are based are provided. The conclusions are appropriately stated and are connected to the original question investigated in that the authors characterize and compare genetic diversity and relatedness in populations of P. gurneyi and P. cafer.

Additional comments

List of suggested corrections
Abstract
In the abstract I advise changing Cape Town to Helderberg Nature Reserve
Introduction
I think the Introduction would flow better if it started at Line 12.
Lines 2 -11 are out of place and should be included later in the introduction after the bird species have been described and introduced. So move to Line 56.
Line 13 and line 53 southern with a small letter ‘s’.
Line 20 Explain the use of the word “abundance” in this line. Sentence is not clear.
Line21 The sentence does not make sense and is incomplete ..used by this species how?
Line 22-23 (Taylor, Peacock & Wanless, 2015; ….)
Line 30 Change the sentence starting on the line to help with English flow:
Taking into account that Proteaceae utilized by ……, and that territory size …..
line 36 to line 37 new paragraph is slightly disjointed ….
Line 56 new paragraph Add lines 1 -11 in here and then go on to the microsatellite mrakers as is from line 56.
Line 57 McFarlane (capital F) and elsewhere in the manuscript.
Methods and materials
Sample collection and DNA extraction
No sample sizes are given except that one individual was caught in Sani Pass. It would help to include the sample numbers for the two populations here
Line 63 µl not ul
Line 64 ….. National Park with a capital letter
Line 65 in Sani Pass
Line 66 Helderberg Nature Reserve,
Line 68 …2003). Permits were obtained from the relevant
Line 83 roadkill not killed
Microsatellite genotyping
Line 97 …used multiplex PCR
Line 98 McFarlane
Line 115 Table 3 not 2
Line 108 …. for both species. (not for either species)
Line 120 not Giving. Should change to …. resulting in an expected….
Line 152 trial runs
Line 159 …with jack-kniffing across loci - small letter j
Results
In Line 178 to 190: Please would the authors explain why there is a reduced sample size amplification for Pro86 and Pro66 in both P. gurneyi and P. cafer. i.e. in Table 2 sample size for P gurnyei is n=49 for all loci except Pro86 (n=24) and Pro66 ( n=31), and for P. cafer n=63 for all loci except Pro86 (n=37) and Pro66 ( n=61),
Line 178 six
Line 196 Sanger with a capital S
Lines 207 The authors write “despite a smaller sample of the former”. This does not make sense to me because P. cafer had the larger sample size of n=63 compared to P. gurneyi having n=49
Line 215 Cervus 3.0.7 (Kalinowski, Taper & Marshall, 2007)

Discussion
Line 231 and 273 McFarlane
Line 235 …has been inferred in bird monitoring….
Line 284 ….may be higher abundance….
Line 287 six
Line 291 This may reflect more first and second order…
Line 298 south is small letter
Line 328 linkage-disequilibrium (small letters)
References
All cited and listed and vice versa
except for Line 347 Altshul et al 1990 is listed but not cited.
Line 412 extra-pair
Line 416 Journal of Primatology

---

## Round 0.2 · accepted · Accept

Dear authors,

Many thanks for the revision, which has taken into account all the issues raised by the reviewers.. Congratulation, we can now accept your ms.

Kind regards

Michael Wink
Academic editor

#